# Dynamic Speed Control of Unmanned Aerial Vehicles for Data Collection under Internet of Things

**DOI:** 10.3390/s18113951

**Published:** 2018-11-15

**Authors:** Qi Pan, Xiangming Wen, Zhaoming Lu, Linpei Li, Wenpeng Jing

**Affiliations:** 1School of Information and Communication Engineering, Beijing University of Posts and Telecommunication, Beijing 100876, China; panqiouc@sina.com (Q.P.); xiangmw@bupt.edu.cn (X.W.); llinp1993@163.com (L.L.); jingwenpeng@bupt.edu.cn (W.J.); 2Beijing Key Laboratory of Network System Architecture and Convergence, Beijing University of Posts and Telecommunications, Beijing 100876, China; 3Beijing Laboratory of Advanced Information Networks, Beijing University of Posts and Telecommunications, Beijing 100876, China

**Keywords:** unmanned aerial vehicles (UAVs), speed control, access congestion, Internet of Things (IoT), data collection, wireless sensor network (WSN)

## Abstract

With the new advancements in flight control and integrated circuit (IC) technology, unmanned aerial vehicles (UAVs) have been widely used in various applications. One of the typical application scenarios is data collection for large-scale and remote sensor devices in the Internet of things (IoT). However, due to the characteristics of massive connections, access collisions in the MAC layer lead to high power consumption for both sensor devices and UAVs, and low efficiency for the data collection. In this paper, a dynamic speed control algorithm for UAVs (DSC-UAV) is proposed to maximize the data collection efficiency, while alleviating the access congestion for the UAV-based base stations. With a cellular network considered for support of the communication between sensor devices and drones, the connection establishment process was analyzed and modeled in detail. In addition, the data collection efficiency is also defined and derived. Based on the analytical models, optimal speed under different sensor device densities is obtained and verified. UAVs can dynamically adjust the speed according to the sensor device density under their coverages to keep high data collection efficiency. Finally, simulation results are also conducted to verify the accuracy of the proposed analytical models and show that the DSC-UAV outperforms others with the highest data collection efficiency, while maintaining a high successful access probability, low average access delay, low block probability, and low collision probability.

## 1. Introduction

After several years’ development, Internet of Things (IoT) has been becoming ubiquitous in our daily lives. Tens of billions of devices with sensing, computing, and communication capabilities will largely change, renovate, and transform the world into an entirely digital earth. All kinds of information are sensed and transmitted to specific servers for various applications, i.e., smart metering, intelligent agriculture, smart grids, and intelligent transportation systems (ITSs). Meanwhile, IoT has also been touted as an economic engine for growth and will generate prominent revenues which exceed $300 billion [1]. The compound annual growth rate (CAGR) will exceed 25 percent for IoT devices and corresponding connectivity segments over the years [2].

In many cases, especially in wireless sensor networks (WSNs), there are no available communication links for pervasive sensor devices in complex harsh environments, such as in disaster districts, mountain regions, desert areas, and so on. Long distances between sensor devices and network infrastructures prevent direct data exchanges. In addition, distributions of sensor devices which need data collection and dissemination are always spatially and temporally related to a low degree. Especially in case of unexpected and temporary events creating hard-to-predict inhomogeneous traffic demand, such as emergency rescue and traffic congestion, deployments of communication infrastructures under these conditions are difficult, high-cost, unprofitable, and unworthy. There are no deployed fixed sink devices nor any communication infrastructures to support data aggregation for such ubiquitous IoT devices. Wireless networks may need additional support to maintain ubiquitous connections.

In order to solve the above issues, balloons, satellites, and piloted aircrafts have been proposed and discussed as the primary platforms applied for data collection and dissemination, bridging communications between sensor devices and a network [3]. It is more efficient to use mobile balloons or satellites to provide temporary communication links for flexible mobility. However, these solutions are always limited by an ultra-high cost, a complicated deployment, weather conditions, and so on.

Fortunately, the explorations of unmanned aerial vehicles (UAVs) as flying wireless communication networks at various altitudes have recently made significant progress to achieve dynamic and adaptive coverage [4,5]. Because UAVs are flexible, portable, inexpensive, and convenient [6], they have been widely applied for various purposes [7], i.e., traffic monitoring, forest fire detection emergency event supervision [6,8], data ferrying [9], communication enhancement [10], data collection [11], 3D city map construction [12,13], and wireless connectivity provisioning (e.g., projects Loon by Google and Internet.org by Facebook) [14]. According to the Federal Aviation Administration, sales of UAVs for commercial purposes are expected to grow from 600,000 in 2016 to 2.7 million by 2020 [15].

Primarily, UAVs have been applied in the military, which were mainly deployed in hostile territory to reduce pilot losses [7]. With the development of the manufacturing industry and material science, drones have gained considerable cost reduction and size minimization. UAVs can be currently categorized into two classes: fixed wing and rotary wing. Each of them has unique characteristics. Generally, fixed-wing UAVs can move with a high speed and can carry heavy payloads. They can be equipped with a greater battery supply for a longer flight time. However, it cannot stay in one fixed point in the air. In contrast, the rotary-wing drones have more flexible mobility and can hover, take off vertically, and land. They allow users to operate in a smaller vicinity with no substantial landing/takeoff area required. The cost behind the flexibility of rotary-wing UAVs is the high decrease in payload and flight time. However, their capacity to hover and perform agile manipulations makes rotary-wing drones well suited to applications such as inspections, where precision manipulation and the ability to maintain a visual on a single target for extended periods of time is required.

Generally, data communication between sensor devices and UAVs can be supported by WLAN (IEEE 802.11 series), ZigBee, Bluetooth Low Energy (BLE), cellular networks, and so on. Owing to the overload burden, high energy consumption, and hidden terminal problems, the IEEE 802.11 technologies based on carrier sense multiple access with collision avoidance (CSMA/CA) are not suitable for UAV-based data collection [16]. For this paper, we have chosen a certain cellular network, i.e., long-term evolution (LTE)/LTE-Advance (LTE-A) network, to gather data from sensor devices due to the higher performance, easy installation, wide coverage, flexible resource management, and reliable quality of service (QoS) guaranteed [17,18]. Cellular networks have been widely used to enhance air-to-ground communications [19,20,21].

Compared to fixed terrestrial base stations and other solutions for data collection, the main advantage of using UAV-based aerial base stations is the ability to quickly and easily move. Furthermore, the high altitude of UAVs can enable line-of-sight (LoS) communication links to the ground users, which makes great promotions for the channel gains. Due to the strong communication links and flexible mobility, UAVs can move toward potential ground users and collect the sensed data with a low power consumption, reducing the energy consumptions of sensor devices.

However, there are still plenty of challenges during the data collection process with UAVs. Obviously, energy efficiency is one of the key problems which have been a hot topic in this field. The energy supply of UAVs is always very limited owing to the constraints on payloads. UAVs are forced to complete data collection missions within a constrained amount of time. What is worse, massive sensor devices lead to substantial signaling congestion for the aerial base station, causing more energy waste. Hence, a cost-effective and energy-efficient solution is urgently needed for UAVs to efficiently complete data collection from tens of thousands of sensor devices.

### 1.1. Related Work

There has been plenty of research focusing on data collection from sensor devices via UAV-based base stations from different aspects, such as trajectory planning, deployment, and autonomous flight control.

In Ref. [22], the convolutional neural network (CNN) was used for feature extraction of necessary information and the deep Q network (DQN) was deployed to support the decision making for UAV control. UAVs automatically cruise in the city and collect most of the required data in the sensing region, while a mobile charging station reaches the target point in the shortest possible time to charge the drones. In Ref. [11], the joint optimization of the sensors’ wake schedules and the UAVs’ trajectories was formulated into a mixed-integer non-convex optimization problem. It was designed to minimize the maximum energy consumption of all sensor devices while ensuring reliable data collections. Mohammad in Ref. [4] investigated the optimal trajectory and deployment of multiple UAVs when they acted as aerial base stations for data collection. UAVs flew to the center of corresponding clusters. Sensor devices were divided via the K-means algorithm for data collection and the trajectories of UAVs were determined by the framework of optimal transport theory. In Ref. [23], a fly-hover-and-communicate protocol was proposed to optimize the UAV’s flying altitude and antenna beamwidth for throughput optimization in three fundamental multiuser communication models. The devices were clustered, and UAVs hovered above each cluster center for data collection. Zanjie in Ref. [24] proposed an optimal dynamic programming based algorithm (DPBA) with the bandwidth and energy allocation jointly considered to maximize the total transmitting rate for data collection via UAVs. However, drones always needed to stay or hover over at fixed places and wait for the completion of data collection, leading to a low efficiency of UAVs. The time needed for data collection was long, and only a few devices could finish the data exchanges within the flight time of the UAVs.

In addition, Yawei in Ref. [25] exploited the wireless power transfer technology to replenish the energy of sensor clusters and developed an efficient data collection scheme for pervasive sensor clusters. Data gathering in rechargeable WSNs was mathematically formulated into an optimization problem for maximizing data collection efficiency. However, the energy supply of UAVs cannot be too high due to the constrained tolerable load. In Ref. [26], the authors proposed a heuristic algorithm for minimizing the energy consumption of data collection with UAVs. With both the location and time constraints considered, it provided a good solution for the data mule scheduling (DMS) problem. In Ref. [27], a cooperative data collection scheme was proposed to gather the sensing data from a WSN via UAVs. Sensor devices utilized UAVs for cooperative data transfer. In Ref. [28], an energy-efficient UAV-based data collection protocol in WSNs was proposed to reduce the energy consumption of sensor devices. Devices were divided into clusters, and an optimal path for UAV was derived for UAVs to collect data from cluster headers.

However, few studies have paid attention to communication collisions on the MAC layer. For typical IoT scenarios, such as the massive machine type communication (mMTC) scenarios of 5G, there are massive sensor devices in the WSN waiting for data upload, which causes severe uplink communication congestion and overload for UAV-based aerial base stations. For example, with cellular base station mounted in UAVs, the ALOHA-based random access procedure (RAP) causes severe collisions, especially in the face of a large number of simultaneous access attempts. Retransmission and back-off processes lead to high energy consumption for both sensor devices and UAVs. Moreover, drones always need to stay or hover at fixed places, waiting for the data upload from devices in the current coverage area. The consumed time is long and the efficiency is also too low with respect to the constrained flight duration of UAVs. Hence, flexible and efficient mechanisms are needed for UAVs to gather data efficiently and simultaneously alleviate the access congestion in the MAC layer.

### 1.2. Contribution & Organization

In this paper, we take the congestion state of a MAC layer into consideration and dynamically adjust UAVs’ speed adaptively for reliable and efficient data collection. As shown in Figure 1, when a ground sensor device enters the communication coverage of drones, it is activated by UAVs and wakes up to establish the network connection via RAP. For simplicity, we assume that UAVs fly straight across the devices and that data collections have to be completed when devices are still in the coverage area. Under the proposed scheme, UAVs dynamically adjust the flight velocity adaptively according to the amount of devices under current communication coverage areas. With congestion status considered, the time for collecting data from sensor devices is significantly decreased, simultaneously reducing the energy consumption of both sensor devices and UAVs. More sensor devices can finish their data transmission within a limited flight duration. The contributions of this paper can be summarized as follows:
A dynamical speed control algorithm for UAVs (DSC-UAVs) is proposed to maximize the data collection efficiency in WSNs. UAVs can adaptively adjust the velocity according to the sensor device density in current coverage areas.In order to well characterize the performance of data transmission from sensor devices to UAVs, detailed mathematical models are presented and analyzed in terms of successful access probability, collision probability, average access delay, and data collection efficiency.Based on the above analytical models, the optimal speed for UAVs under different sensor device densities is derived to maximize data collection efficiency, which is verified via computer calculations.Simulations have also been conducted to evaluate the accuracy of the proposed analytical models and verify the effectiveness of the DSC-UAV mechanism.


The rest of this paper is organized as follows. Section 2 describes details of the system model for the data collection with UAV-based aerial base stations, including the RAP for sensor devices. The analytical models for the connection establishments and optimal speed derivation are presented in Section 3. Simulations are described and analyzed in Section 4 and Section 5 concludes the paper at last.

## 2. System Model

In this section, the system model for data collection in WSN via UAV-based aerial base stations is described in detail. Data are collected to UAVs via cellular network technologies and temporarily stored in the UAVs. Once UAVs fly back to the ground destination, the data from sensor devices can be sent/utilized for various subsequent purposes.

Assume there are *M* sensor devices distributed in a given area, waiting for data collection, and one fixed-wing or rotary-wing drone is assigned to fly along this area for data collection. The UAVs’ trajectory is assumed to be a straight line for simplicity. In addition, we assume that the density of devices along the drone’s trajectory can be denoted as λ per square-meter, and the width of this area can be regarded as 2∗k, where *k* is assumed to be far less than the coverage radius for simplicity, as shown in Figure 2.

In addition, we assume that the UAV keeps the fixed altitude during its flight for simplicity and then the available communication coverage radius on the ground can be represented as *r*. When sensor devices appear in the available communication coverage, they will be activated via the corresponding air-interface activation mechanisms of UAVs. Sensor devices prepare to establish the uplink communication links. After the four-step handshaking, connection between devices and UAVs is established for data transmission. Due to the small-data characteristic of IoT devices, we assume that, once the connection is established, the data transmission would be finished in the same time.

Currently a typical delegate of cellular networks, LTE/LTE-A technologies are utilized for UAV-based aerial base stations [5]. Under an LTE/LTE-A network, the basic time-frequency resource unit is the resource block (RB) and the minimum unit of bandwidth is 180 MHz, consisting of 6 RBs. The time resource is divided into frames of fixed length and each frame consists of 10 subframes [29]. One subframe is further divided into two time slots. The RAP can only be started in fixed time slots, according to the pre-allocated PRACH configuration index. These fixed slots’ duration, $T_{RA\_REP}$, is the interval between two consecutive random access slots (RASs) [2]. We assume that UAVs fly across this area with a speed, *v*, and the number of new arrivals, ΔM, can also be derived as follows:
(1)$$\Delta M=v*T_{RA\_REP}*\lambda *2k.$$


Device *A*, as shown in Figure 2, needs to finish the data transmission before the UAV leaves the available communication coverage area. In other words, devices need to complete the RAP within the coverage area of the drones. Otherwise, sensor devices have to abandon their data transmission due to the limited retransmission and available communication time. Hence, we need to ensure that
(2)$$\begin{array}{c} T_{delay}\leq \frac{2L}{v}\\ L=\sqrt{r^{2}-k^{2}} \end{array}$$
where *L* indicates the minimum available coverage distance for sensor devices, and $T_{delay}$ is the time needed for connection establishments to UAVs. The access delay of RAP mainly depends on the number of current access attempts, the back-off window size, the available radio access resources, etc. More practically, we assume that all sensor devices immediately wake up to transmit their sensed data to UAVs once receiving the broadcast activation messages.

### 2.1. Definition of Data Collection Efficiency

In order to reveal the efficiency of aerial base stations, the data collection efficiency needs to be defined and analyzed in this subsection.

The definition of data collection efficiency should reflect the capability of UAVs to handle data collection from massive sensor devices. The flight duration of UAVs is assumed to be fixed for power-constrained drones. The number of sensor devices which successfully complete data transmission to UAVs within the flight duration can be considered to show the data collection efficiency. Hence, the data collection efficiency, *U*, can be defined as follows:
(3)$$U=\frac{M_{s}}{T}$$
where $M_{s}$ indicates the number of devices whose data collection has been successfully completed, and *T* denotes the RASs of the flight duration for the UAVs.

### 2.2. Random Access Procedure under a Cellular Network

In this subsection, details about how the connection between sensor devices and UAVs is established is described as shown in Figure 3.

When a sensor device is at its first initial access, not synchronized, close to handover or after a radio link failure, a separate physical random access channel (PRACH) is provided for connection establishment. The RAP is classified into two types: contention-free and contention-based RAP. The first is mainly deployed where connection to the network has already existed, e.g., handover among different base stations. Otherwise, devices employ the contention-based RAP for establishing a network connection. For energy-saving in WSNs, the sensor devices always stay in sleep mode unless activated by UAVs. For simplicity, the RAP represents the contention-based RAP for the rest of this paper.

Via broadcast System Information Block 2 (SIB2) messages from the flying Evolved Node B (eNodeB), devices learn about the details of access channels, including the PRACH configuration index, the PRACH frequency offset, and the available number of preamble sequences. Details of the four-step contention-based RAP are shown in Figure 3.
**Preamble Transmission**: Devices randomly choose one from the available preamble sequences, *R*, and transmit it to the eNodeB as an access request. The preambles contain RA radio network temporary identity (RA-RNTI) and preamble configuration index information. When more devices choose one same preamble sequence, the eNodeB cannot decode this message. As a result, collisions occur. After transmitting the preamble, devices wait for the random access response (RAR) and the number of retransmissions, *n*, are also increased one by one.**Random Access Response**: The RAR is sent by eNodeB on the physical downlink sharing channel (PDSCH). In addition, via the RA-RNTI, the transmission time between devices and the eNodeB can be calculated, and the RAR conveys a timing alignment (TA) instruction for determining the subsequent transmission synchronization, an initial uplink resource grant, and a temporary cell RNTI (C-RNTI), which is mandatory in the contention-based RACH procedure. A back-off indicator, $W_{BO}$, is also included. In addition, if RAR messages cannot be received within the RAR time window, this access request is deemed a failure and needs to be retransmitted. However, if multiple devices choose the same preamble sequence, they receive the same RAR message if eNodeB cannot detect the collision.**Layer 2/Layer 3 (L2/L3) Message**: This is the first scheduled message in the physical uplink sharing channel (PUSCH) and makes use of the hybrid automatic repeat request (HARQ). It conveys a temporary C-RNTI allocated in Step 2 for identity recognition, RRC connection request, tracking area update (TAU), and so on. Collided devices will encounter conflicts in the same uplink time-frequency resources when transmitting their L2/L3 messages. As a result, the eNodeB cannot decode their network connection requests because interferences and devices restart the RAP after reaching the maximum number of HARQ retransmissions.**Contention Resolution Message**: After receiving Message 3, the eNodeB returns acknowledgment (ACK), conveying the temporary C-RNTI for successful devices via the physical downlink sharing channel (PDSCH). Devices that can find their IDs in Message 4 continue data transmission. Otherwise, devices go back to sleep mode, perform the back-off algorithm, and re-initiate a new RAP until the limited retransmission times are reached.


## 3. UAV-Based Data Collection Analysis and Modeling

### 3.1. Modeling of Connection Establishments

In this subsection, we model RAP for collection establishments in terms of the successful access probability, $P_{s}$, the average access delay, $T_{delay}$, and data collection efficiency, *U*. As UAVs need to collect data from massive sensor devices, severe collisions occur. Many sensor devices fail in their data transmission because of the limited retransmission or because they do not have enough time for data transmission due to the high speed of UAVs.

The number of access attempts in the first RAS, $M_{1}$, can be denoted as follows [30]:
(4)$$\begin{array}{cc} M_{1} & =\Delta M\\ M_{1,s} & =M_{1}e^{\frac{-R}{M_{1}}}\\ M_{1,f} & =M_{1}-M_{1,s} \end{array}$$
where $M_{1,s}$ and $M_{1,f}$ denote the number of successful devices and collided devices, respectively, in the first RAS. Collided devices restart their transmission again after the random back-off procedure.

The number of retransmissions is not fixed, as the time when devices are in the coverage of UAV-based aerial base stations may be so small that the maximum retransmission cannot be reached. As the flow chart shows in Figure 4, before retransmission, sensor devices firstly check if they are still under the coverage of UAVs. When UAVs are flying with a speed of *v*, the maximum retransmission, $n_{max,v}$, can be derived as
(5)$$\begin{array}{c} T\left[n\right]=\left(W_{RAR}+T_{RAR}\right)*n+\left(n-1\right)*W_{BO}/2\\ T\left[n\right]+W_{BO}/2=\left(W_{RAR}+T_{RAR}+W_{BO}/2\right)*n\\ T_{th,v}=\frac{2*r}{v}\\ n_{max,v}=min\left(N_{PT_{max}},\left\lfloor \frac{T_{th,v}+W_{BO}/2}{W_{RAR}+T_{RAR}+W_{BO}/2}\right\rfloor \right) \end{array}$$
where $T_{RAR}$ and $W_{RAR}$ respectively indicate the time needed for the transmitted preambles to be detected by the aerial base station and size of the RAR window. $N_{PT_{max}}$ represents the original maximum number of retransmission for RAP under the cellular LTE/LTE-A network. T[n] indicates the average time needed for *n* retransmission, and $T_{th,v}$ is the time threshold for sensor devices to finish their data transmission when the velocity of UAVs is *v*, which is also the time duration when devices are under the service coverage. The maximum retransmission for the velocity of UAVs, $n_{max,v}$, can thus be derived as shown in the above equation.

Hence, the number of access attempts in the *i*th RAS, $M_{i}$, can be denoted as the sum of devices with different retransmissions, as follows:
(6)$$M_{i}=\sum_{n=1}^{n_{max,v}}M_{i,n}$$
where $M_{i,n}$ expresses the number of sensor devices executing their *n*th retransmission in the *i*th RAS.

The number of sensor devices, $M_{i}$, includes not only the newly arrived access attempts, ΔM, but also collided devices, which restart their retransmission in the *i*th RAS, $M_{p,i}$. We can derive
(7)$$\begin{array}{c} Len=\lceil \frac{W_{BO}+T_{RAR}+W_{RAR}}{T_{RA\_REP}}\rceil \\ M_{p,i}=\vec{\alpha }\cdot \vec{M_{r}}\\ \vec{\alpha }=\left[\alpha_{0},\alpha_{1},\cdots ,\alpha_{Len-1}\right]\\ \vec{M_{r}}={\left[M_{f,i-1},M_{f,i-2},\cdots ,M_{f,i-Len}\right]}^{T} \end{array}$$
where $\vec{\alpha }$ is the probability distribution of the RAS when previous collided devices restart RAP. The element of $\vec{\alpha }$, $\alpha_{k}$, is the probability that collided devices will restart their transmissions after *k* RASs. $\vec{M_{r}}$ indicates the vector of previous collided devices which may start the next retransmission in the current RAS. The length of vectors, $\vec{\alpha }$ and $\vec{M_{r}}$, mainly depends on the maximum number of RASs between the collision and retransmission for devices, Len.

Hence, we can derive the number of sensor devices in the *i*th RAS ($0\leq j\leq n_{max,v}$):
(8)$$\begin{array}{c} M_{i}=\Delta M+M_{p,i}\\ M_{i,1}=\Delta M\\ M_{i,1,s}=M_{i,1}\cdot e^{-\frac{M_{i}}{R}}\\ M_{i,1,f}=M_{i}-M_{i,1,s}\\ M_{i,2}=M_{f,i-1,1}\cdot \alpha_{0}+M_{f,i-2,1}\cdot \alpha_{1}+\cdots M_{f,i-Len,1}\cdot \alpha_{Len-1}\\ M_{i,2,s}=M_{i,2}\cdot e^{-\frac{M_{i}}{R}}\\ M_{i,2,f}=M_{i,2}-M_{i,2,s}\\ \cdots \;\;\cdots \;\;\cdots \\ M_{i,j}=M_{f,i-1,j-1}\cdot \alpha_{0}+\cdots +M_{f,i-Len,j-1}\cdot \alpha_{Len-1}\\ M_{i,j,s}=M_{i,j}\cdot e^{-\frac{M_{i}}{R}}\\ M_{i,j,f}=M_{i,j}-M_{i,j,s} \end{array}$$
where $M_{i,j,s}$ and $M_{i,j,f}$ respectively denote the number of successful and collided devices on their *j*th transmission in the *i*th RAS.

We can then easily derive the successful access probability, $P_{s}$, and the collision probability, $P_{b}$, as follows: The successful access probability can be defined as the ratio between the number of successful connection-established devices and total arrived access attempts during this time period. The collision probability is defined as the ratio between the number of collided devices and all the access attempts in all RASs during UAVs’ flights.

Based on the analytical model above, we can also obtain the average access delay for successful devices to establish the connections to UAVs. The delay can be defined as the time duration between the first preamble transmission and the successful complements of RAP [2]. Devices which are on their *j*th preamble retransmission have experienced j−1 access collisions and random back-off periods. We can then derive the average access delay for each device as follows:
(9)$$T_{delay}=\frac{\sum_{i=1}^{T}\sum_{j=1}^{i-1}M_{i,j,s}\cdot T\left[j\right]}{\sum_{i=1}^{T}M_{i,s}}.$$


Hence, the data collection efficiency can be derived as follows:
(10)$$U=\frac{M_{s}}{T}=\frac{\sum_{i=1}^{T}\sum_{j=1}^{n_{max,v}}M_{i,j,s}}{T}.$$


### 3.2. Dynamic Speed Control for UAVs

Assume that the UAVs fly directly across one area for data collection as shown in Figure 2. As sensor devices are deployed for environment sensing or other event monitoring, the density distribution of sensor devices are known for the users. Hence, we assume that the UAVs with GPS information can easily learn the sensor device density in the current area. The UAV velocity or sensor device density in this area thus have a great influence on the data collection efficiency without a doubt.

If a UAV flies with a high speed, or if the density of a sensor device is high for the current area, the number of newly arrived access attempts per RAS, ΔM, is so large that the flying radio access network ultimately suffers heavy congestion and overload. Plenty of devices come across collisions and keep retransmitting their access requests. As a result, most devices cannot send data to UAVs, leading to high energy consumption and low data collection efficiency.

When UAVs fly with a low speed, the arrival rate is low and there are no collisions during the connection establishment process. UAVs can only collect data from a small number of sensor devices owing to the low arrival rate. The whole number of successful sensor devices within the flight duration of UAVs is too small, leading to low access resource utilization and a low data collection efficiency.

Hence, an optimal speed of UAVs under different sensor device densities is needed to maintain an optimal arrival rate, alleviating network congestion and simultaneously maximizing data collection efficiency. The optimization problem can be expressed as follows.
(11)$$\begin{array}{cc} \max_{v}\; & U\\ s.t.\; & v>0\\ & T_{delay}<L/2v\\ & M_{i},M_{i,j,s},M_{i,j,f},M_{p,i}\in Z^{\geq 0} \end{array}$$
where $Z^{\geq 0}$ indicates non-negative integers.

Assume the access process for massive sensor devices reaches a steady state where the successful access number of sensor devices per RAS, $\bar{M_{s}}$, is steady [31]. Obviously, more successful devices than $\bar{M_{s}}$ means more access attempts in this RAS, causing severer collisions and congestion. More failed devices start to retransmit, increasing the subsequent number of access attempts per RAS and finally leading to serious congestion where only a few devices can finish the RAP. Hence, we can conclude that $\bar{M_{s}}$ is unique and alsothe maximum number of successful devices for relatively long time periods. Therefore, we can derive
(12)$$\begin{array}{c} \bar{M}=\left(\Delta M+\bar{M_{f}}\right)\\ \bar{M_{s}}=\Delta M\\ \bar{p_{s}}=\frac{\bar{M_{s}}}{\bar{M_{f}}+\bar{M_{s}}}=e^{-\frac{\bar{M_{f}}+\bar{M_{s}}}{R}} \end{array}$$
where $\bar{M_{f}}$ and $\bar{p_{s}}$ indicate the collided devices in each RAS and successful access probability, respectively.

Easily we can obtain the probability distribution of average access delay under the steady state as shown in Table 1.

Based on the above Table 1, the average access delay under steady status can be derived:
(13)$$\bar{T_{avg}}=\sum_{i=1}^{n_{max,v}}{\left(1-\bar{p_{s}}\right)}^{k-1}\cdot \bar{p_{s}}\cdot \left(kT_{b}+\frac{\left(k-1\right)W_{BO}}{2}\right)$$
where $T_{b}=W_{RAR}+T_{RAR}$ for simplicity. We can derive that the data collection efficiency under a steady state is a function of the successful access probability:
(14)$$\bar{U}=\frac{\bar{M_{s}}}{\bar{T_{avg}}}=f\left(\bar{p_{s}}\right).$$


Hence, we can easily derive the optimal $M^{*}$ under different sensor device densities to maximize data collection efficiency. This optimal velocity results from the steady state of access process, which can be regarded as a result of the steady state. In addition, the optimal speed can also be acquired based on the previous analytical models (i.e., Equation (11)) of data collection via computer calculations. Via the previous analytical models, the data collection efficiency under different sensor device velocities and densities can be obtained via computation. The optimal speed of UAVs that maximizes the data collection efficiency can also be figured out, which can be regarded as results of computations. The optimal velocity resulting from the steady state is also verified by comparison with the results from the previous analytical models through computations, as shown in Figure 5.

The numerical results shown in Figure 5 show that results derived from the steady state consistently match with the optimal results from the computer calculations. When the sensor device density is low, drones need to maintain a high speed to maximize the resource efficiency, and UAVs need to slow down in order to alleviate the congestion and overload when the sensor device density continues to increase. The velocity of UAVs can then be adjusted according to the density of the sensor devices, maximizing the collection efficiency.

## 4. Simulations

In this section, the details of simulations are elaborated for evaluating the performance of our proposed DSC scheme. The analytical results are also verified by comparison with the simulated access process. The DSC scheme of UAVs for data collection is also verified, which can adaptively adjust the speed according to different amounts of access attempts from sensor devices under the UAVs’ coverage area.

The simulations were conducted with MATLAB software. Proposed analytical mathematical models are verified in terms of the successful access probability, block probability, average access delay, and data collection efficiency. Table 2 states the parameter settings in the simulations. For simplicity, the channel is assumed to be ideal without fading or loss in order to reflect the effects of access scheduling in the MAC layer. In other words, transmission failure only results from access collisions or transfinite retransmissions.

The analytical models for the data collection period are validated from the perspective of successful access probability, average access delay, block probability, data collection efficiency, and collision probability, as shown in Figure 6. The density of devices in this area is assumed to be 100 per square-meter, and the data collection procedure between devices and UAVs is simulated for comparison to verify the accuracy of the analytical models (i.e., see Section 3.1). The simulated results are the performances of the RACH procedure under different velocities of UAVs. Apparently, the proposed mathematical models accurately characterize the simulated access procedure for UAVs to collect data from massive sensor devices with minor deviations.

When the UAVs fly across this field with a low speed, the successful access probability is as high as 1, the block probability and collision probability is 0, and the average access delay also remains low. The number of newly arrived access attempts is so small that the limited radio access resources are entirely sufficient for the connection establishments and data collection. With the velocity of UAVs increasing, the collision probability and average access delay starts to increase, while the successful access probability and block probability remains steady. This is because more access attempts per RAS arrive due to the higher velocity in the UAV. Collisions occur during the data collection establishment procedure, and devices successfully finish the RACH procedure with a few back-off and retransmission, which accounts for the increase in average time delay. However, when the velocity of UAVs continues to increase to about 20 m/s, as shown in Figure 6, the performances worsen. The successful access probability encounters a sharp decrease. The block probability, collision probability, and average access delay also increase dramatically. Due to the high speed of UAVs, the number of access attempts per RAS becomes larger and the limited radio resources become insufficient for data transmission to UAVs. The finite retransmission and back-off mechanism cannot ensure successful connection establishment due to severe collisions. Hence, some sensor devices encounter severe collisions and are blocked with high collision probability, decreasing the successful access probability and increasing the block probability. Moreover, the average access delay increases as successful sensor devices require several retransmission and back-off periods. Finally, when UAVs fly with a higher speed of more than 25 m/s, as shown in Figure 6, the successful access probability decreases to 0. The block probability and the average collision probability maintain at 1. The average access delay encounters a sudden reduction. This is because a higher velocity of UAVs indicates a greater amount of access attempts per RAS. The collisions are so serious that sensor devices still cannot establish the network connection to UAVs after finite retransmission and back-off waiting periods. Very few sensor devices successfully finish RAP occasionally, which accounts for the decrease in the average access delay.

In addition, the data collection efficiency is also derived and presented in Figure 6d. Easily we can see that the simulated results are completely consistent with the analytical results. There is a maximum value for the data collection efficiency when the speed of UAVs is about 15 m/s. With this point of speed, the number of sensors that can finish the data transmission to UAVs per RAS can be maximized. On the one hand, when the speed is lower, the successful access probability can be as high as 1, and the average access delay also maintains a low value. However, the number of arrived access attempts per RAS is also lower with low data collection efficiency. On the other hand, more arrived access attempts per RAS lead to serious collisions and heavy congestion, which leads to a lower number of successfully accessed sensor devices with a longer access delay.

In order to maximize data collection efficiency, UAVs need to adaptively adjust their speed according to the sensor device density under their coverage area. Based on the analytical models verified above, the optimal speed of UAVs under different sensor device densities can be derived and evaluated as shown in Figure 5. According to the sensor device density in the area, drones can adaptively adjust their speed to maximize data collection efficiency. The number of newly arrived access attempts in each RAS then remain relatively steady and appropriate in order to avoid the access congestion and simultaneously maximize data collection efficiency.

For the sake of evaluations and verifications of the proposed dynamic speed control algorithm, simulations are conducted. Assume that UAV drones fly directly across one place with a high sensor device density. The distribution of sensor devices in the area of 500 m in length is shown in Figure 7. Performances of UAVs with fixed velocities of 5 m/s, 10 m/s, and 20 m/s are also simulated for comparisons from the perspective of successful access probability, average access delay, block probability, data collection efficiency, and average collision probability as shown in Figure 8.

Firstly, when the sensor device density is low, all schemes behave well. However, data collection efficiencies of the other three schemes with fixed velocity maintain very low values. This is because the number of arrived access attempts per RAS is rare due to the low density under the current area. Limited radio access resources are completely sufficient for the small number of arrived access attempts with few collisions. The radio resources are not taken full use of, leading to low data collection efficiency. This can be treated as a waste of the energy for power-constrained drones. The UAV-based aerial base station is supposed to handle more sensor devices’ data transmission. Gradually, the data collection efficiency for the scheme with higher velocity reaches its maximum point when the distance is about 40 m and λ = 60/m^2^ for velocity = 20 m/s. The high speed of UAV drones means more arrival attempts per RASand higher data collection efficiency, as shown in Figure 6d. The number of successfully accessed sensor devices per RAS is maximized. The access resources are fully utilized. With the sensor device density increases, the scheme with higher velocity sees a sharp decline in its performance due to serious collisions, causing the access network to suffer heavy congestion. As a result, UAVs cannot complete data collection. Similar conditions also occur for the other schemes with smaller fixed velocities.

When the UAV drone flies above the place with the highest sensor device density, the performances of other three schemes with fixed velocities all become the worst for successful access probability, average access delay, block probability, and data collection efficiency, as shown in Figure 8.

However, the performance of the DSC-UAV scheme is always the best compared to the other three schemes with fixed velocities. The successful access probability always remains as high as 1 and the block probability also remains as 0 all the time. The average access delay and collision probability also keep relatively low. They are only a little higher than the three compared schemes when the sensor device density is low. This is because the data collection efficiency is too low for schemes with fixed velocity when the density of devices is also low. There are too small number of access attempts per RAS, leading to radio access resource wastage and low data collection efficiency. Therefore, the DSC-UAV scheme increases the velocity, causing more arrival attempts per RAS. Slight collisions then occur, and the collision probability of the proposed scheme is higher than others. Retransmission and back-off procedures are needed for these collisions, which could account for the higher average access delay.

To further reveal the effectiveness of the proposed DSC-UAV scheme, the number of successful sensor devices and the successful access probability during the flight across this area is presented in Figure 9. Easily we can see that the successful access probability is 1, and all sensor devices’ data can be successfully collected via the proposed DSC scheme. Other schemes with fixed velocities can only collect a part of these data with low successful probabilities.

In summary, via simulation, the proposed analytical models accurately characterize the performances of data collection for sensor devices by UAVs. In addition, based on the analytical modes, the proposed DSC scheme for UAVs is verified to maximize data collection efficiency in comparison with the schemes with fixed velocities.

## 5. Conclusions

In order to maximize the data collection efficiency for UAVs, a dynamic speed control scheme for UAVs is proposed to adaptively cope with different sensor device densities. Firstly, the data transmission procedure between devices and UAVs is modeled and analyzed in detail. The number of successfully accessed devices per second is defined as the data collection efficiency for UAVs. Based on the analytical models, the optimal velocity under different sensor device densities can be derived to maximize data collection efficiency. Finally, simulation results show the accuracy of the proposed analytical models and verify the effectiveness of the DSC-UAV scheme. In the future, the channel status will be considered as a key factor in speed control due to the uniqueness of LoS communication links.

## Figures and Tables

**Figure 1 sensors-18-03951-f001:**
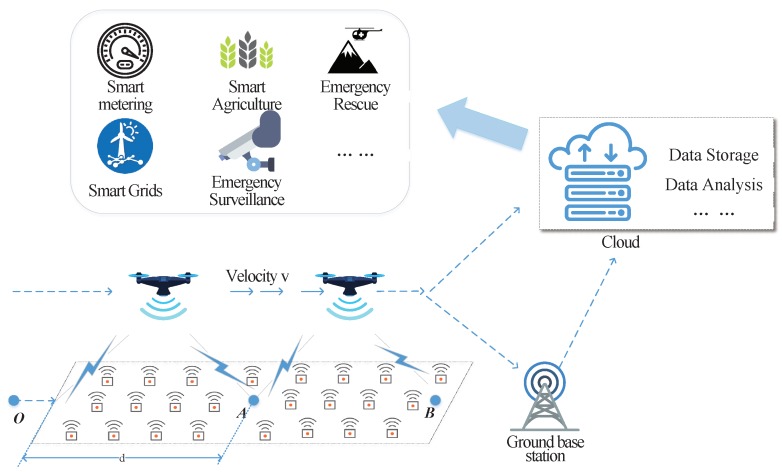
Data collections via UAVs.

**Figure 2 sensors-18-03951-f002:**
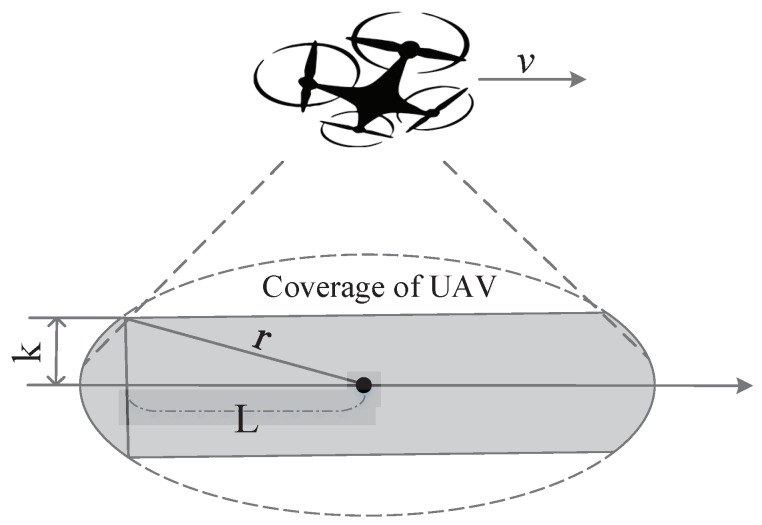
Coverage analysis of UAV-based aerial base stations.

**Figure 3 sensors-18-03951-f003:**
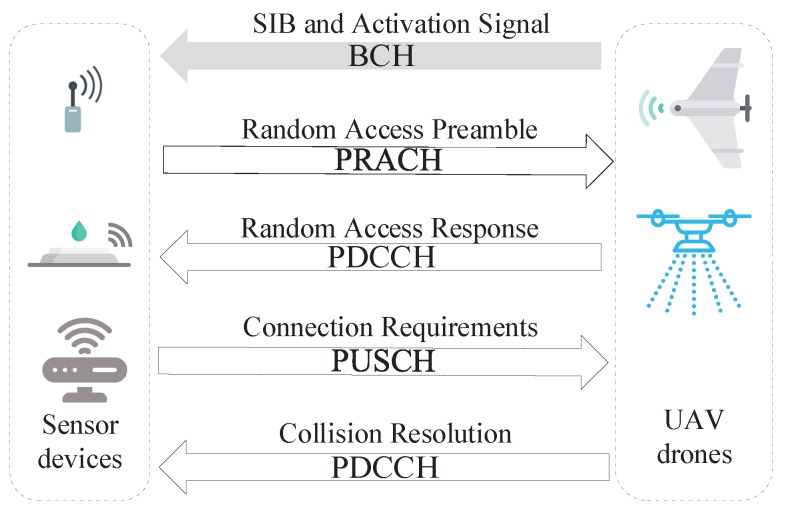
Random access procedure under a cellular network.

**Figure 4 sensors-18-03951-f004:**
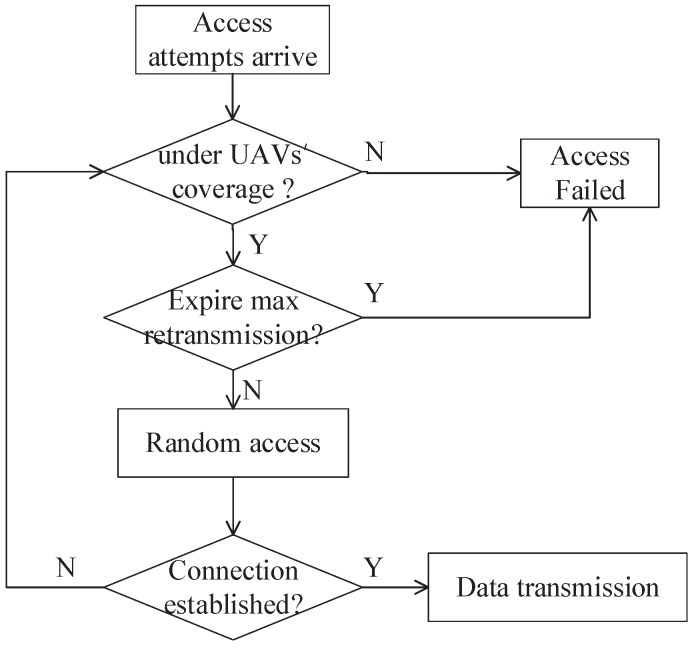
Connection establishment between sensor devices and UAVs.

**Figure 5 sensors-18-03951-f005:**
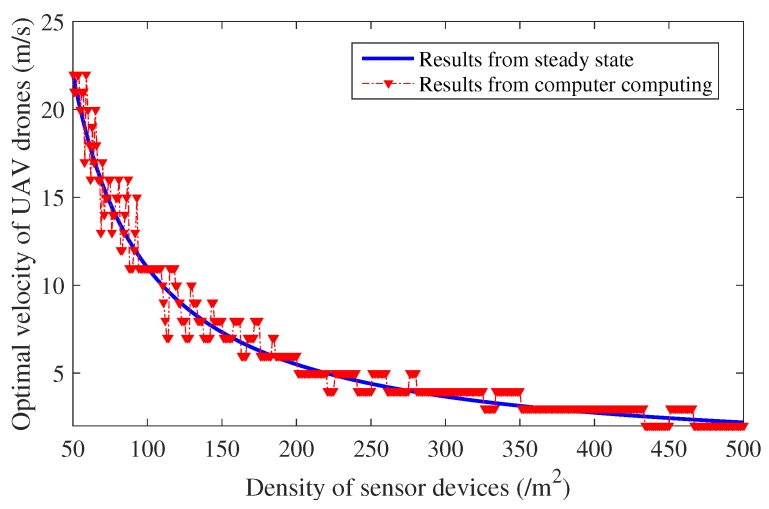
Optimal speed for UAVs under different sensor device densities.

**Figure 6 sensors-18-03951-f006:**
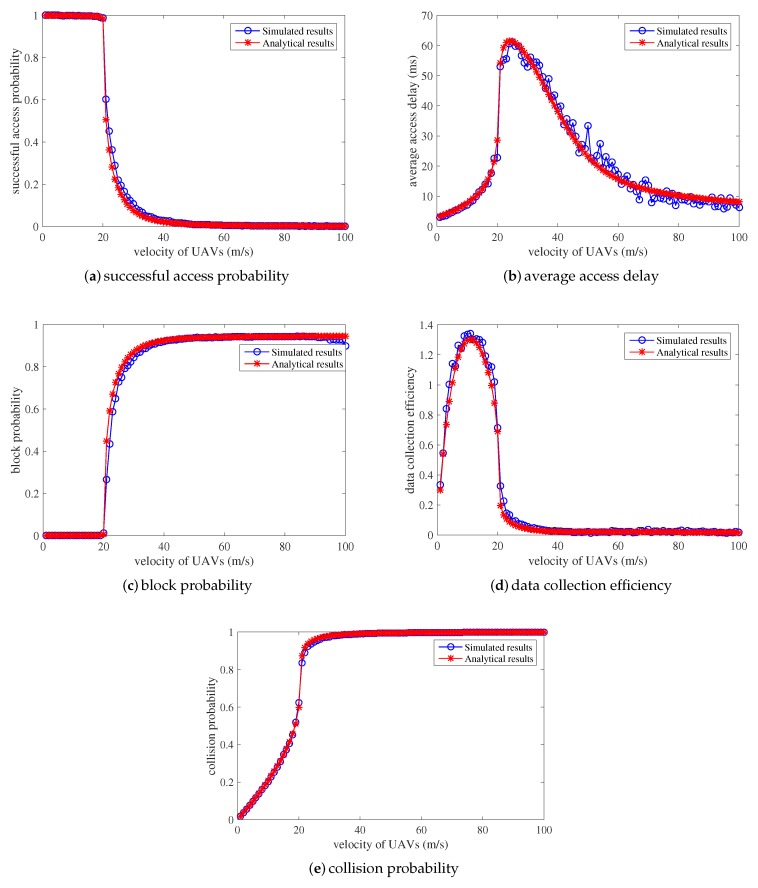
Verification of proposed analytical models when λ=100.

**Figure 7 sensors-18-03951-f007:**
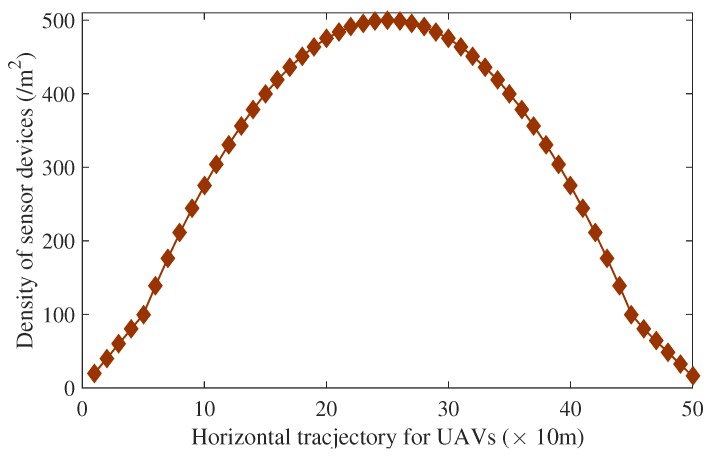
Density distribution of sensor devices in this area.

**Figure 8 sensors-18-03951-f008:**
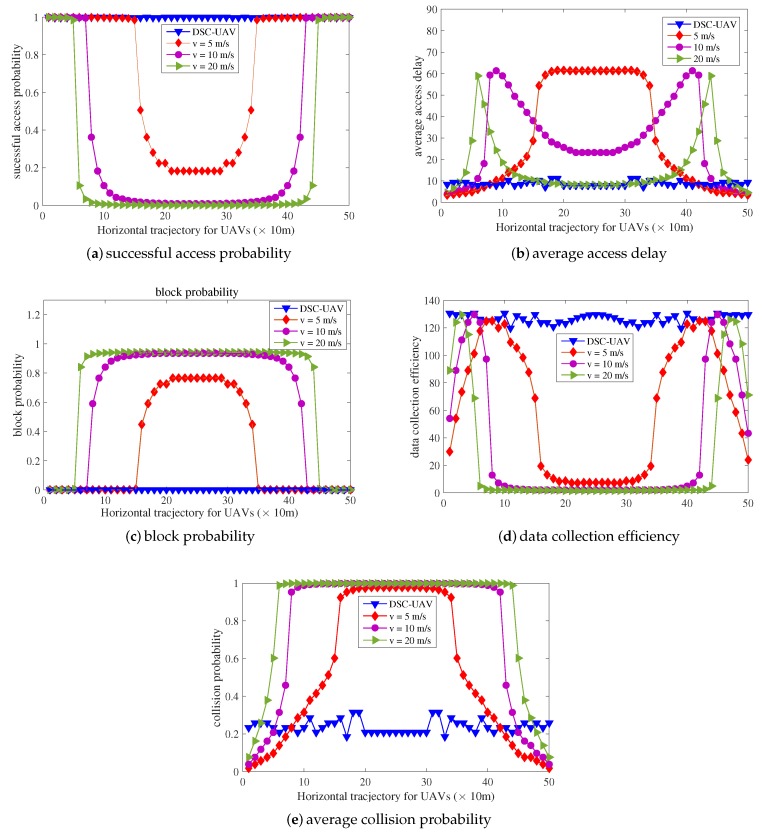
Verification of the proposed dynamic speed control scheme.

**Figure 9 sensors-18-03951-f009:**
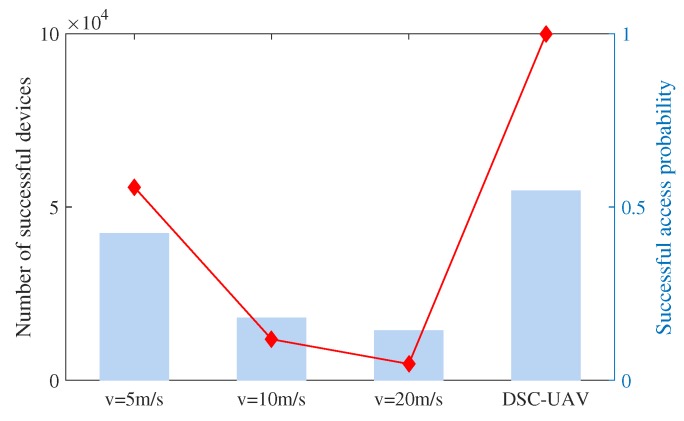
Successful number and probability.

**Table 1 sensors-18-03951-t001:** $\bar{T_{avg}}$ and probability for different retransmissions.

Retrans	1	2	…	k	…
Prob	$\bar{p_{s}}$	$\left(1-\bar{p_{s}}\right)\bar{p_{s}}$	…	${\left(1-\bar{p_{s}}\right)}^{k-1}\bar{p_{s}}$	…
Delay	$T_{b}$	$2T_{b}+\frac{W_{BO}}{2}$	…	$kT_{b}+\frac{\left(k-1\right)W_{BO}}{2}$	…

**Table 2 sensors-18-03951-t002:** Simulation parameter settings.

Parameters	Values
PRACH configuration Index	6
Total number of preambles	54
Time duration of LTE subframe	1 ms
Maximum retransmission times under LTE	10
Back-off indicator of LTE	20
Density of sensor devices	0 to 500
Radius of UAVs’ coverage	30 m

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
