# Peer review of "Dynamic Speed Control of Unmanned Aerial Vehicles for Data Collection under Internet of Things"

_sensors, 2018, doi:10.3390/s18113951_

Reviewer 1 Report

In this paper, a dynamic speed control algorithm for UAVs (DSC-UAV) is proposed to maximize the data collection efficiency, while alleviating the access congestion for the UAV-based base stations.

The paper is well written and easy to follow, and the paper deserve to be published after addressing the following comments.

The reviewer is interested   know to collision probability of the RACH and it can be improved with how much is more than UAV is taking the same trajectory.

In addition, the reviewer is interested to analyze the velocity and the altitude of UAV on the connection establishment  

In addition, some similar works that are recently proposed [a-e] deserve to be discussed in your introduction to strengthen the paper and give potential readers complete idea regarding the progress in this hot research area.

Finally another iteration of proofread is required to fix all typos in this work.

[a] Y. A. Nijsure, G. Kaddoum, G. Gagnon, F. Gagnon, C. Yuen and R. Mahapatra, "Adaptive Air-to-Ground Secure Communication System Based on ADS-B and Wide-Area Multilateration," in IEEE Transactions on Vehicular Technology, vol. 65, no. 5, pp. 3150-3165, May 2016

[b] Y. Li and L. Cai, "UAV-Assisted Dynamic Coverage in a Heterogeneous Cellular System," in IEEE Network, vol. 31, no. 4, pp. 56-61, July-August 2017.

[c] Y. Nijsure, M. F. A. Ahmed, G. Kaddoum, G. Gagnon and F. Gagnon, "WSN-UAV Monitoring System with Collaborative Beamforming and ADS-B Based Multilateration," 2016 IEEE 83rd Vehicular Technology Conference (VTC Spring), Nanjing, 2016, pp. 1-5.

[d] N. Tadayon, G. Kaddoum and R. Noumeir, "Inflight Broadband Connectivity Using Cellular Networks," in IEEE Access, vol. 4, pp. 1595-1606, 2016.

[e] H. He, S. Zhang, Y. Zeng and R. Zhang, "Joint Altitude and Beamwidth Optimization for UAV-Enabled Multiuser Communications," in IEEE Communications Letters, vol. 22, no. 2, pp. 344-347, Feb. 2018.

Reviewer 2 Report

Please take into conssideration the followings:

Line 84: explicit first mentioned "MTC"

Line 138 to140: explain how the drone learns (or knows) about the density of the sensor devices in the current overflight area.

Line 231: (below): fill the right reference.

Section 4. Simulation: First, it should be made clear what is the analytical model (using the relationships of the mathematical model) and the data with which it was validated. Second, the simulated model should be described. In fact, it is not clearly shown which are the two distinct models (analytical and simulated) that compare (obtaining the graphical results in Fig. 6).

Line 339 versus Line 162: mention to "lambda" is in meters or in square meters?

The profile of the (variable) speed provided by the DSC UAV scheme should be graphically represented and commented for certain simulation scenarios at least!

Usually, the analytical model of an optimization problem should define an objective function for optimization or an expression, giving the optimal value according to the considered criteria (average access delay and data collection efficiency). Which is your explicit function?

Author Response

Round  2

Reviewer 1 Report

The paper is ready for publicaton